# Coco Monoethanolamide Surfactant as a Sustainable Corrosion Inhibitor for Mild Steel: Theoretical and Experimental Investigations

**DOI:** 10.3390/molecules28041581

**Published:** 2023-02-07

**Authors:** Richika Ganjoo, Shveta Sharma, Praveen K. Sharma, O. Dagdag, Avni Berisha, Eno E. Ebenso, Ashish Kumar, Chandrabhan Verma

**Affiliations:** 1Department of Chemistry, School of Chemical Engineering and Physical Sciences, Lovely Professional University, Phagwara 144402, India; 2Centre for Materials Science, College of Science, Engineering and Technology, University of South Africa, Johannesburg 1710, South Africa; 3Department of Chemistry, Faculty of Natural and Mathematics Science, University of Prishtina, 10000 Prishtina, Kosovo; 4NCE, Department of Science and Technology, Government of Bihar, Patna 803108, India; 5Interdisciplinary Research Center for Advanced Materials, King Fahd University of Petroleum and Minerals, Dhahran 31261, Saudi Arabia

**Keywords:** non-ionic surfactant, coco mono ethanol amide, corrosion inhibition, electrochemical and theoretical techniques

## Abstract

Recent studies indicate that surfactants are a relatively new and effective class of corrosion inhibitors that almost entirely meet the criteria for a chemical to be used as an aqueous phase corrosion inhibitor. They possess the ideal hydrophilicity to hydrophobicity ratio, which is crucial for effective interfacial interactions. In this study, a coconut-based non-ionic surfactant, namely, coco monoethanolamide (CMEA), was investigated for corrosion inhibition behaviour against mild steel (MS) in 1 M HCl employing the experimental and computational techniques. The surface morphology was studied employing the scanning electron microscope (SEM), atomic force microscope (AFM), and contact measurements. The critical micelle concentration (CMC) was evaluated to be 0.556 mM and the surface tension corresponding to the CMC was 65.28 mN/m. CMEA manifests the best inhibition efficiency (η%) of 99.01% at 0.6163 mM (at 60 °C). CMEA performs as a mixed-type inhibitor and its adsorption at the MS/1 M HCl interface followed the Langmuir isotherm. The theoretical findings from density functional theory (DFT), Monte Carlo (MC), and molecular dynamics (MD) simulations accorded with the experimental findings. The MC simulation’s assessment of CMEA’s high adsorption energy (−185 Kcal/mol) proved that the CMEA efficiently and spontaneously adsorbs at the interface.

## 1. Introduction

The persistent and spontaneous deterioration of metal structures caused by chemical and/or electrochemical interactions with environmental components is known as corrosion. [1,2,3]. Failures due to corrosion have a significant impact on several businesses, particularly those reliant on the petroleum industry where metallic materials are widely used [4,5]. The corrosion-related failures have a negative impact on the productivity and ongoing operations of these businesses. Because of the corrosion failures, various industrial and metallurgy operations, including descaling, electrolyte-based cleaning, pickling, and acidification, cause a significant loss of metallic elements throughout the surface treatment [6,7]. Numerous economic and safety issues are brought on by corrosion failure. Corrosion currently costs the world economy more than USD 2.5 trillion annually [8]. Corrosion failures have been blamed for several accidents, particularly in the petrochemical sector. These initiatives aim to cut the cost of corrosion by 15 to 35% which is equivalent to USD 375 billion to USD 875 billion, respectively [8,9]. Due to growing urbanization and industrialisation, the negative effects of corrosion, such as safety and financial penalties, are anticipated to increase. Therefore, depending on the type of metal and electrolytes, corrosion experts have devised numerous attempts. Every effort has advantages and disadvantages unique to it. Nevertheless, it has been proven that one of the best and most widely used techniques in aqueous electrolytes is the employment of organic molecules [10,11,12,13]. Numerous organic compound families, particularly heterocycle ones, are widely used in corrosion prevention [14,15,16]. But they face a problem with low solubility, particularly for those with aromatic rings (s). Surfactants, on the other hand, are a relatively recent class of corrosion inhibitors that have nearly all the necessary characteristics for a material to be used as an aqueous phase corrosion inhibitor [17,18,19,20]. They have the proper ratio of hydrophilicity (polar functional groups) to hydrophobicity (hydrocarbon chains), which is primarily required for powerful interfacial interactions [17,18,19,20].

Surfactants have an amphiphilic structure consisting of a polar head and a nonpolar tail. They could be bola form, Gemini-type, zwitterionic, anionic, cationic, nonionic, or anionic, depending on the type of head. Surfactants are commonly used for lubrication, anti-wear, degreasing, anti-scaling, and acid picking of iron and steel, as well as mineral ore processing and acidification of oil wells. They are also extensively employed to avoid the corrosion of metals and alloys [19,21,22,23,24]. Surfactants are a particular family of organic inhibitors that adsorb on metal surfaces, offering effective metal corrosion protection [25,26,27]. The study and synthesis of surfactants (non-ionic, anionic, cationic, and zwitterionic) as inhibitors in diverse media have been the focus of many researchers [28,29,30,31,32,33,34,35,36]. In addition to being ecologically friendly and sustainable, non-ionic surfactants also have the potential to reduce corrosion. They have centres with a high electron density (N, S, O, P, π electrons and aromatic ring), enabling them to adsorb on metallic surfaces and protecting them from corrosive environments [37,38,39,40]. Some green corrosion inhibitors, according to reports, are, however, costly or only work well when administered in high concentrations. Non-ionic surfactants are reported to exhibit significant inhibitory efficiency for the corrosion of metals in a variety of corrosive conditions and to have noticeably lower critical micelle concentrations (CMCs) than equivalent ionic surfactants [19,38,41,42]. Undoubtedly, exploring ecological and inexpensive corrosion inhibitors for metal corrosion inhibition is significant. Coco mono ethanol amide (CMEA) is a coconut-based surfactant, derived from the fatty acids from coconut oil and monoethanolamine (MEA) and it is widely used in cosmetic products and causes no harm to marine life or human safety. The structure of CMEA shows that the nonpolar hydrophobic moiety is a long carbon chain, and the polar hydrophilic moiety of the compound comprises readily protonated nitrogen and oxygen atoms that provide the surfactant with a remarkable anti-corrosion potential. This research aims to examine the mechanism of CMEA’s inhibition and its effectiveness for MS in 1 M HCl. The findings of weight loss (WL), electrochemical impedance spectroscope (EIS), and potentiodynamic polarization (PDP) to investigate the surfactant’s inhibitory performance are presented in Section 3.2 and Section 3.3, respectively, and to examine the surface characteristic of mild steel (MS), scanning electron microscope (SEM), atomic force microscope (AFM), and contact angle examination were carried out; the discussions are provided in Section 3.4. To validate the experimental results and to comprehend the connection between the inhibitor structure and MS, theoretical studies are also performed.

## 2. Result and Discussion

### 2.1. CMC Determination

The surface tension (ST) measurements were done at various doses of CMEA and the CMC was calculated to be 0.556 mM by plotting ST against concentration (Figure 1). The surface tension is reduced with the increase in dose and then becomes constant. The point where the linearly dependent area’s regression straight line intersects with the straight line traversing the region of constant surface tension yields the CMC. A surface tension reading of 65.28 mN/m was recorded at the CMC.

### 2.2. WL Studies

#### 2.2.1. Impact of Temperature and Concentration

The WL evaluations were done at different concentrations at 30–60 °C to comprehend the mode of adsorption (physical or chemical). The results of the WL experiments are exhibited in Table 1. It can be concluded that the values for inhibitory effectiveness (WL%) and surface coverage (θ) rise along with the concentration of CMEA. The highest η%_WL_ was obtained at a dose of 0.6163 mM. With the rise in temperature from 30 °C to 60 °C, the inhibition effectiveness increased and was highest at 60 °C and 0.6163 mM concentration, with an inhibition efficiency of 99.01%, and no increase in efficiency was seen beyond this concentration, which is accredited to the MS’s active areas being fully saturated. The increase in the inhibition effectiveness at higher temperatures is ascribed to enhancement in the energy of the adsorbate molecules to cross the activation energy barrier easily and leads to chemisorption. The corrosion rate, ν (mg.cm^2^ h^−1^), is dependent on both the temperature and concentration.

The WL experiment results were used to plot the graphs between η%_WL_ vs. temperature and corrosion rate vs. concentration (Figure 2). The graphs demonstrate how temperature influences inhibition efficiency, and also the necessity of the corrosion rate.

#### 2.2.2. Activation Parameters Studies

The activation parameters like activation entropy (∆S*), activation enthalpy (∆H*), and activation energy (*E*_a_) were studied for the optimum concentration of 0.6163 mM at 30, 40, 50, and 60 °C to further examine the inhibition properties of CMEA and their dependence on the corrosion rate (ν). The activation parameters were calculated by the Arrhenius and Eyring equations given below [33,34,43,44]:(1)$$\log\left(\nu \right)=\log A-E_{a}/2.303RT$$
(2)$$\nu =\frac{RT}{Nh}\exp\left(\frac{\Delta S^{*}}{R}\right)\exp\left(-\frac{\Delta H^{*}}{RT}\right)$$
where v, *T*, *h*, *N*, *A*, and *E*_a_ denote the corrosion rate, temperature, Planck constant, Avogadro number, Arrhenius pre-exponential factor, and corrosion activation energy, respectively.

The log v vs. 1/T (Equation (5)) and log(v/T) vs. 1/T (Equation (6)) were plotted to attain *E*_a_ and ∆S* and ∆H* values. The activation parameters obtained for 0.6163 mM CMEA are displayed in Table 2. The ∆H* value is positive, indicating endothermic adsorption, which means the adsorption increases with a temperature increase. The negative value of ∆S* designates a decrease in the entropy attributed to the CMEA adsorption. The Arrhenius and Eyring plots for 0.6163 mM concentration of CMEA are exhibited in Figure 3.

#### 2.2.3. Thermodynamic and Isotherm Studies

The adsorption isotherms at different temperatures (30–60 °C) were studied to comprehend the metal surface and the CMEA interaction. The surface coverage (θ) data acquired from the WL experiments were applied to various isotherms. The data fitted accurately into the Langmuir adsorption isotherm (LAI; R^2^~1). The LAI equation is given below [45,46]:(3)$$\frac{C_{\mathrm{inh}}}{\theta }=C_{\mathrm{inh}}+\frac{1}{K_{\mathrm{ads}}}$$
where the equilibrium adsorption constant is denoted by $K_{\mathrm{ads}}$ and whose values were derived from the straight line’s intercept, which was obtained by plotting C_inh_/θ against C_inh_. The LAI at different temperatures is displayed in Figure 4.

The value of *K*_ads_ attained at varying doses of CMEA are accessible in Table 3. The high k_ads_ values with the temperature rise recommend that the binding energy of the inhibitor increase with the temperature.

The $\Delta G_{ads}^{0}$ values were derived from the *K*_ads_ using Equation (8) [47,48]:(4)$$K_{\mathrm{ads}}=\left(\frac{1}{55.5}\right)\exp\left(\frac{-{\Delta G}^{\circ }\mathrm{ads}}{\mathrm{RT}}\right)$$

The physical symbols have their standard meaning [45,46]. From Table 3, it can be seen that the $\Delta G_{ads}^{0}$ were negative, suggesting the spontaneous reaction of the inhibitors being adsorbed on the MS, and the values are between −30 and −40 KJ mol^−1^ and hence, signify mixed adsorption, i.e., both physical (around −20 kJ mol^−1^) and chemical adsorption (>−40 kJ mol^−1^); and the adsorption tends to go towards chemisorption with increasing temperature.

The heat of the adsorption ($\Delta H_{ads}^{0}$) value obtained from the Van ’t Hoff equation (Equation (9)) displayed in Table 3 was positive, indicating an endothermic reaction [49,50]:(5)$$\log K_{\mathrm{ads}}=\frac{-\Delta H^{\circ }\mathrm{ads}}{2.303RT}+\;\mathrm{constant}$$

$\Delta H_{ads}^{0}$ was obtained by the slope of a straight line by plotting log k_ads_ vs. 1/T. The $\Delta S_{ads}^{0}$ values were obtained by substituting the other thermodynamic values in the basic thermodynamic equation given below [51,52]:(6)$$\Delta G_{ads}^{\circ }=\Delta H_{ads}^{\circ }-T\Delta S_{ads}^{\circ }$$

The adsorption entropy ($\Delta S_{ads}^{0}$) values decreased with the temperature, indicating an enhancement in the adsorption of the inhibitor. The $\Delta S_{ads}^{0}$ values for the CMEA adsorption were calculated to be positive with no discernible change at various temperatures.

### 2.3. Electrochemical Studies

#### 2.3.1. Open Circuit Potential (OCP) Studies

For MS, the OCP was measured in 1 M HCl concerning time for varying concentrations of CMEA, and the OCP for a blank solution was evaluated to be 0.432 V. The solution was kept unstirred for 1 h and after that, the OCP was run for 500 s and a straight line parallel to the time axis exhibited the achievement of steady-state potential. The OCP values obtained for different concentrations with respect to time, as shown in Figure 5. The OCP values shifted for the inhibited solution to higher negative values, demonstrating that the inhibitor primarily adsorbs on the cathodic sites.

#### 2.3.2. Potentiodynamic Polarization (PDP) Studies

The voltage is applied between the MS and SCE in a potentiodynamic polarisation (PDP) experiment, and the current is monitored. The data obtained from the PDP experiment were used to plot the log (i) vs. *E*, and the Tafel curves were obtained for MS in 1 M HCl at 30 °C for variable doses of CMEA, as shown in Figure 6. From Figure 6, it can be concluded that the Tafel slopes at different inhibitor concentrations follow a regular trend with decreasing corrosion current density (j_corr_). The shift towards the cathodic direction demonstrates that the inhibition was predominantly cathodic and inhibited the hydrogen evolution reaction. The PDP parameters, including *E*_corr_ (corrosion potential), j_corr_ (current density), *b*_a_ (anodic Tafel slope), and *b*_c_ (cathodic Tafel slope), are achieved by using the corrosion rate analysis in the nova 2.1.4 programme to extrapolate the cathodic and anodic regions of the Tafel curves. The results are tabulated in Table 4.

As may be noticed from the findings displayed in Table 4, with the rise in the inhibitor dose, the j_corr_ values dropped from 1028 to 26.632 µA cm^−2^ for blank to 0.6163 mM CMEA because an adsorbed layer of CMEA developed on the MS surface. The *b*_a_ and *b*_c_ both showed a small change in their values, with increasing concentration demonstrating that the corrosion kinetics was not affected by the inhibitor’s presence, but the change was predominant in the *b*_c_, suggesting the inhibition phenomena to be cathodic controlled, which was further supported by the *E*_corr_ values being shifted towards a more negative direction.

The *E*_corr_ values showed a shift of <85 mv, signifying CMEA to be a mixed-type inhibitor, and it controlled both the hydrogen evolution (cathodic) and metal dissolution (anodic) reaction. The inhibition efficiency η_PDP_% improved from 60.09 to 97.11%. The highest inhibition efficiency was obtained for 0.6163 mM close to the CMC of the surfactant, and beyond that, no more increase in the efficiency was seen, owing to the surfactant’s adsorbed layer being saturated.

#### 2.3.3. Electrochemical Impedance Spectroscopy (EIS) Studies

The EIS experiments were executed at 30 °C to comprehend the kinetic and mechanistic behaviour of the electrochemical system under consideration. The Nyquist plots were constructed using the information from the EIS findings (Figure 7) and the Bode phase angle and frequency plots (Figure 8) for blank and inhibited test solutions. It was evident from the Nyquist curves that the width of the capacitive loops expanded as the inhibitor doses rose, attributed to an improvement in the surface’s absorption of the inhibitor film. The depression in the loops was because of the non-uniformity and coarseness of the metal surface and frequency dispersion, and also suggested that the MS corrosion was controlled by the charge transfer process. The shape of the impedance curves did not change, implying that there was no modification to the corrosion mechanism with the inhibitor addition. Employing the nova 2.1.4 software, the most appropriate electrochemical equivalent circuit was used to simulate the EIS findings as revealed in Figure 7, consisting of a series connection between the solution resistance (R_s_) and these impedance components, together with a parallel connection between the constant phase element (CPE) and the polarisation resistance (R_p_). The electrochemical characteristics that were evaluated from the equivalent circuit are outlined in Table 5.

To take into consideration the surface inhomogeneity caused by surface roughness, displacements, defects, and the dispersion of active areas, the CPE is utilised instead of a pure capacitor. Mathematically, the expression below is used to calculate the impedance of a CPE [53,54]:(7)$$Z_{CPE}=\frac{1}{Y_{o}{\left(j\omega \right)}^{n}}$$
where j denotes an imaginary number, Y_0_ and n (ranges from 0 to 1) signify the constant phase element (CPE) and phase shift, respectively, and ω represents the angular frequency. The CPE exponent provides a measurement of the surface’s heterogeneity and has values in the range of 0 to 1.

Double-layer capacitance (C_dl_) values were evaluated employing Equation (12) [55,56]:(8)$$C_{\mathrm{dl}}=\frac{1}{2\pi f_{maxR_{p}}}$$
where *f*_max_ is the frequency at which the impedance’s imaginary component has the highest value.

Table 5 demonstrates that *R*_s_ values were quite low as compared to the charge transfer resistance (*R*_ct_) values, which demonstrated that the corrosion mechanism was primarily regulated by the transfer of electrons between the MS and the defensive layer and the resistance of the specimen to oxidation during the application of an external potential. The acquired values of R_s_ indicate that the solution conductivity is decreased by the introduction of the CMEA, which relates to increased blockage at the metal surface’s active sites. The obtained values of *R*_s_ are greater for the solution that contains an inhibitor. It is evident that *R*_ct_ enhanced as the inhibitor dosage increased, which validated that the inhibitor’s adsorption layer had greater corrosion resistance. The *R*_ct_ values increased from 21.45 ohm.cm^2^ to 758.53 ohm.cm^2^ for blank and 0.6163 mM inhibitor concentration. In addition, the reduction in C_dl_ value was seen because of the enhancement in the width of the double layer or drop in the local dielectric constant as a result of inhibitor adsorption on the MS surface. This is steady with the Helmholtz model, as shown by the subsequent equation [57,58]:(9)$$C_{dl}=\frac{\varepsilon \varepsilon_{0}}{\delta }S$$
where the protective layer’s dielectric constant and the permittivity of free space (8.854 × 10^−14^ F cm^−1^) is denoted by ε and $\varepsilon_{0}$, δ represents the protective layer’s width and S stands for the electrode’s surface area. In order to anticipate the dissolution process, phase shift (*n*) was considered as an indicator. The steady n values demonstrated that the charge transfer mechanism controlled the dissolution without and with various concentrations of CMEA.

The Bode plots obtained by plotting the EIS results are presented in Figure 8 (Bode phase angle plots) and Figure 8 (Bode impedance plot). In the Bode phase angle plot, the phase angle was plotted against frequency to recognize the impact of frequency on the corrosion protection and at the surface/solution interface, and it was observed that there is just one time constant associated with the creation of an electric double layer. A basic understanding of the inhibitory activity of the inhibitors may be acquired from the phase angle at high frequencies. The magnitude of the capacitive electrochemical behaviour increases with increasing negative phase angle value. The ideal capacitor has a maximum phase angle of 90° at the intermediate frequency. The phase angle value is lower in the absence of CMEA than it is in its presence (Figure 8). The rise in the maximum phase angle with increasing inhibitor doses facilitates the additional inhibitor molecules’ adsorption on the MS surface, hence decreasing the metal dissolution rate and enhancing its protective qualities. The Bode impedance graphs (Figure 8) demonstrate that absolute impedance values increased at low frequencies with an increase in inhibitor doses. This specifies that the defensive abilities of the inhibitor increase as its concentration increases.

### 2.4. Surface Examination

#### 2.4.1. SEM Examination

The formation of an adsorbed layer of surfactant was confirmed with the help of the SEM. The MS coupons were dipped for 6 h in 1 M HCl without and with 0.6163 mM concentration of CMEA. The results of the SEM analysis are displayed in Figure 9. The polished MS sample showed a smooth surface (Figure 9 a) without any pits and cracks. After 6-h immersion in 1 M HCl, the MS surface was corroded immensely and pits and cracks were seen, as shown in Figure 9. When the MS coupon was dipped in 1 M HCl, having 0.6163 mM CMEA for 6 h, the MS surface showed a clear smooth surface with reduction of surface roughness owing to the formation of an inhibitor defensive barrier and a reduced effect of the corrosive medium.

#### 2.4.2. AFM Examination

To further study the three-dimensional topology of mild steel, AFM analysis was performed for the metal coupons dipped in 1 M HCl for 6 h without and with 0.6163 mM CMEA. The 3-D and 2-D images obtained are displayed in Figure 10. A polished surface (Figure 10a,b) with average roughness (*R*_a_) of 7.31 nm (Table 6) was attained for the reference MS sample. The *R*_a_ of freely corroded MS surface without CMEA after 6-h immersion time was 327 nm (Figure 11c,d). The 3-D image (Figure 10c) clearly shows a corroded surface with roughness and irregularities in the topography. In the presence of an inhibitor in an acidic medium, the least surface roughness was obtained and a much smoother surface was obtained (Figure 10e,f).

The average roughness values and root mean square values were calculated with the help of nano scope analysis; 1.5 software was used. The inhibited MS sample exhibited the least average surface roughness of 20.4 nm, which was very low as compared to the blank as a consequence of the defensive film of CMEA formed on the MS.

#### 2.4.3. Contact Angle Investigation

The hydrophobicity of an inhibitor layer was assessed using the contact angle (CA) experiment. After being submerged in the corrosive medium at 30 °C for 6 h, the surface contact angles of MS steel specimens were evaluated. If a water drop is more likely to adhere to itself than to a particular surface, that surface is termed hydrophobic, and the water drop will bead up with a CA larger than 90°. If a water drop prefers to adhere to a surface more than it adheres to itself, the surface is hydrophilic, and the drop will have a contact angle smaller than 90°. Figure 11 displays the water contact angle on MS surfaces. The contact angle of the polished MS surface was measured for reference and was found to be 100.4° and demonstrating hydrophobicity because of an angle greater than 90°. The surface contact angle of the uninhibited specimen is only 61.40° owing to the existence of hydrophilic corrosion products (Figure 11b). After adding 0.6163 mM of CMEA, the contact angle increases to 92.6° (Figure 11c). The metal surface becomes hydrophobic once the corrosion inhibitor is added, avoiding the corrosive medium’s damaging the metal. The increase in the contact angle approves the defensive layer of CMEA formed on the metal surface.

### 2.5. Theoretical Analyses

#### 2.5.1. DFT Results

DFT studies help in considering the mode of the inhibitory effect of corrosion compounds; hence, they are extensively used in corrosion research [22,59]. Utilizing these computational methods is essential for comprehending the association between molecular structures and protection capabilities [60]. DFT simulations have recently been carried out to determine the adsorption mechanism based on the inhibitor molecule’s structure, the metal, and the aggressive medium [61,62].

Figure 12 displays the optimised molecular structures, and the frontier molecular orbitals (HOMO, LUMO, and ESP) of the CMEA inhibitor, while Table 7 includes the quantum chemical indices such as *E*_HOMO_, *E*_LUMO_, ∆E, I, A, µ, χ, η, σ, ∆N, and ∆E_back-donation_ derived from DFT calculations for the molecule. According to molecular orbital theory, the orbitals that promote chemical reactions are the HOMO and LUMO, which define the lowest energy transition. A bond is formed by the donation and acceptance of electrons from HOMO to LUMO, respectively, where the donor and the acceptor act as a Lewis base and Lewis acid correspondingly. Hence, higher values of *E*_LUMO_ enhance the donor ability and lower values of ELUMO indicate higher accepter ability. The calculated *E*_HOMO_ and *E*_LUMO_ values of CMEA are −7.203 and 0.414 eV, correspondingly, which indicate the facile electron transfer from HOMO of the inhibitor to LUMO of the Fe atom and support the chemisorption. The effectiveness of the electron transfer, chemical stability, easier polarization, and effective polarization of the molecule was further supported by the less energy gap (∆E) as seen in Table 7.

It is recognised that the molecule is successfully adsorbed on the metal surface if it has a contact angle of ~0° owing to the greater contact area covered by the inhibitor. As seen, in Figure 12 the CMEA molecule has a nearly planar structure; hence, efficient adsorption is assumed due to the enhanced surface area covered in adsorption. The ESP of the CMEA was studied to identify the electron density of the molecule and electrophilic and nucleophilic sites in CMEA [63,64].

Figure 12 reveals that electrophilic reactivity is related to positive (blue) areas of ESP, while nucleophilic reactivity is associated with negative (red) regions, And that the electron-richest regions are primarily near conjugated bonds and heteroatoms, which promotes the creation of a chelate on the MS surface by electron transfer from amide and hydroxy groups to the “d” orbital of the Fe atom and the chemical adsorption of CMEA, which results in the development of a covalent coordination bond [65,66].

Figure 12 shows that HOMO and LUMO are mostly distributed around amide groups in CMEA molecules, suggesting the favoured locations for electrophilic attack by metal cations on nitrogen and oxygen atoms [67,68,69]. Another crucial concept is the dipole moment (μ), which measures the bond polarity and distribution of electrons in the molecule. The dipole moment (μ) helps in determining the inhibition effects of organic inhibitors [70,71]. In the present study, the dipole moment is 7.459 D for CMEA (Table 7), which is higher than μH_2_O (1.88 D), and indicates robust dipole–dipole interaction in between the MS and inhibitor and also, the molecules of the inhibitor. Higher CMEA molecule aggregation on the MS surface is favoured by the high dipole moment by the electronic force and results in a significant inhibitory efficiency.

The charge transfer from the inhibitor to the Fe and back donation was favoured energetically owing to the value of ∆E_back-donation_ < 0 and hardness > 0. The electron transport from the CMEA to the metal surface was further reinforced by the value of ∆N > 0 and the transfer of electrons from the metal to the inhibitor occurs when ∆N < 0 [72]. Additionally, Lukovit’s finding states that the inhibitor’s inhibitory efficacy steadily rises if ∆N < 3.6 [73]. In the present experiment, N values are > 0 and < 3.6, demonstrating the transport of electrons from the CMEA to the MS surface. This supports CMEA’s very acceptable inhibitory performance demonstrated via electrochemical and weight loss tests.

#### 2.5.2. MC and MD Simulations

The MC and MD simulations were done to examine the adsorption energy distributions for the CMEA and the adsorption phenomena of the inhibitor, respectively. The MC and MD simulation results of CMEA adsorption in the modelled corrosive medium on the Fe surface are displayed in Figure 13. MD is often seen as a more accurate portrayal of the dynamics of adsorption. The CMEA inhibitor takes on a flattened shape on the surface of MS after several hundred ps of NVT simulation and is heavily adsorbed into the Fe surface (Figure 13).

CMEA had a rigid structure with a –C=O π bond, and electrons for the vacant iron “d” orbital would come from heteroatoms containing lone pairs of electrons to make coordinate bonds. Consequently, the molecule adsorbs on the Fe (110) surface in a planner manner which could maximize the interaction between CMEA and the MS surface, as displayed in Figure 13.

The following Equation (14) is used to determine the adsorption energy (Eads) for the investigated inhibitors on the surface of Fe (110) [74]:(10)$$E_{ads}=E_{total}-\left[E_{surface+water}+E_{inhibitor}\right]$$
where *E*_total_ is the system’s overall energy as a consequence of the CMEA and the metal surface interaction; *E*_surface + water_ is the energy of the Fe (110) surface in combination with water molecules before adsorption; and *E*_Inhibitor_ is the free energy of the CMEA, respectively.

In MC simulation, a huge number of arbitrarily chosen mixtures of molecules and ions are generated in a simulation box. The several configurations are examined until the system reaches its energy equilibrium, which is shown by a smoothing of the mean average energy profile. Figure 14a shows a characteristic energy profile for CMEA adsorption on Fe (110) in a vacuum, which comprises average total energy, total energy, electrostatic, van der Waals, and intramolecular energy. The results of MC simulations show that the CMEA interacts effectively with the Fe surface (Figure 14b), with −185.85 kcal/mol adsorption energy (selected from the max value of P[E]) [75,76].

The length of the bond between the Iron and the atoms of CMEA was calculated using the radial distribution function (RDF) analysis of the MD trajectory. By evaluating bond length values, the various bond types that were formed were recognized [77,78]. Peaks in the RDF graph that arise at certain distances from the metal surface provide information about the type of adsorption activity occurring on the metal [67,79,80]. When the peak is present between 1 and 3.5 Å, it depicts the chemisorption mechanism, and at distances higher than 3.5 Å, it signifies physisorption. The O and A atom’s RDF peak values are shown in Figure 15, and the inhibitors are less than 3.5 Å away from the Fe surface (Figure 15), indicating that the interaction between Fe (110) and inhibitor is mostly chemisorption [65,81].

### 2.6. The Inhibition Mechanism

It is well documented that organic compounds including surfactants become effective by getting adsorbed on the metal surface. In the present investigation, the outcomes of different analyses suggest that CMEA adsorbs on the metal surface and builds a defensive layer. The adsorption of CMEA on the MS surface in 1 M HCl can be described by employing the physiochemisorption mode, just as with conventional organic corrosion inhibitors. As seen in Figure 16b, protonation of the heteroatoms of CMEA in the forms of >C=O (carbonyl), -OH (hydroxyl), and >NH (2°-amine) can occur easily in an acidic solution. Therefore, the CMEA can exit in its mon-, di-, and/or tri-protonated (cationic) forms in 1 M HCl medium. As opposed to that, the build-up of counterions causes the metallic surface to develop a negative charge (hydroxide and chloride ions) [82,83]. Through physisorption, these diametrically opposed charged moieties are attracted to one another. Therefore, interactions between CMEA and the MS surface in an acidic solution may begin with physisorption.

However, the outcomes of the present study suggest that chemisorption is the true mechanism of CMEA adsorption. This could result from heteroatoms deprotonating as they approach a metallic surface by taking in electrons. Therefore, through a process known as donation or transfer, heteroatoms (N and O) move their unshared electron pairs to metallic d-orbitals. Additionally, because metals (in this example, iron) are already electron-rich species, this form of charge transfer results in an interelectronic repulsion state [82,83]. This causes the iron to donate its additional electron through a mechanism known as retro- or back-donation to the empty p-orbitals of C, O, and N. There is a noticeable relationship between the extent of the donation and the extent of the retrodonation, and this relationship is known as synergism [82,83]. The mechanism of corrosion inhibition and modes of CMEA adsorption on the MS surface in 1 M HCl are presented in Figure 16.

## 3. Experimental

### 3.1. Specimen, Reagents and Materials

The MS coupons were acquired from JK steel companies (located in Punjab, India) with chemical composition C-2.83%, Si-0.38%, P-0.02%, V-0.27%, Mn-0.47%, and the rest was Fe. Coco mono ethanol amide (Figure 17) was purchased from BLD pharma, India. Coco monoethanolamide (CMEA) is a non-ionic and waxy surfactant. CMEA contains the minimum amount of free amine and is typically used in formulations that are pH-sensitive. CMEA is well recognized for its well-known moisturizer effect on finished goods. This product is also known to contain and make excellent use of a superb emulsifier, thickening, and wetting ingredient. Additionally, it has significant oil solubility and good oil emulsifier qualities. CMEA can also be used as a stabilizer and foam booster. CMEA modifies the structure of foam to produce richer, denser foam. It is excellent for shampoo, shaving cream, and liquid soap formulations when used as a viscosity controller. The electrolyte (HCl 37%), acetone, and ethanol were procured from Sigma Aldrich. The coupons to be used for weight loss measurements were carved into dimensions (4.5 cm × 4 cm × 0.2 cm) and were grazed with emery paper of diverse grades (180 to 2200) to attain a smooth and glass surface and cleaned with purified water and dried before undergoing the experiment. For the electrochemical experiment, a 5.5 cm long MS rod having a 1 cm^2^ uncovered area was applied.

### 3.2. Surface Tension Measurement

The surface tensions of CMEA at critical micelle concentration (CMC) were measured using a Kruss K9 tensiometer with a thermostable vessel holder at 25 ± 0.1 °C using the Platinum ring detachment technique.

### 3.3. Weight Loss Investigations

The WL study was done at four different temperatures (30–60 °C) on MS coupons having an 18 cm^2^ area. The coupons were grazed with emery papers to remove any dirt and scales present and then splashed with distilled water, acetone, and ethanol and weighed after drying. The samples after weighing were submerged in the test solution contained in six beakers, i.e., 150 mL of 1 M HCl (Blank) and five with different inhibitor concentrations 0.2054 mM, 0.3081 mM, 0.4109 mM, 0.5136 mM, 0.6163 mM added to the test solution for 6 h at 30–60 °C temperature. The temperature was regulated with the help of an oven. After immersion in the different experimental solutions for 6 h, the MS specimens were taken out of the beakers and then cleaned and washed with water acetone and dried with a hot air blower. The weight loss of the coupons was measured accurately thereafter. The corrosion rate (ν, mg cm^2^ h^−1^) and inhibition effectiveness ($\eta_{\mathrm{wL}}\%)$ was evaluated by the Equations (1) and (2), respectively [58,84]:(11)$$\nu =\frac{W_{0}-W}{At}$$
(12)$$\eta_{\mathrm{WL}}\left(\%\right)=\frac{\nu_{0}-\nu }{\nu_{0}}\times 100$$
where $W_{0}$ and W denote the WL earlier and after dipping in a corrosive medium, respectively; $\nu_{0}$ and ν signify the corrosion rate of the blank specimen and the corrosion rate in the inhibitor’s presence, correspondingly.

### 3.4. Electrochemical Techniques

A typical three-electrode 1 L corrosion cell of Autolab PGSTAT204 equipment, 302N model with a FRA32M impedance analyser, was employed to perform the PDP and EIS experiments. An inbuilt nova software 2.1.4 version was utilized to calculate the electrochemical parameters. The test solution having an MS rod (1 cm^2^ exposed area) dipped in it as a working electrode was kept for 1 h initially to attain steady-state equilibrium potential, i.e., open circuit potential (OCP). The unstirred test solution’s OCP was then scrutinized as a function of time till a straight line parallel to the X-axis was attained that validated the achievement of steady-state potential. After the OCP was attained, the PDP experiment was run under a potential range of −0.25 to + 0.25 V at a scan rate of 0.001 V/s. The inhibition efficacy $\;\left(\eta_{PDP}\%\right)$ of the PDP experiment was evaluated by the Equation (3) as follows [85]:(13)$$\eta_{PDP}\left(\%\right)=\frac{j_{\mathrm{corr}\;}^{0}-j_{\mathrm{corr}\;}}{j_{\mathrm{corr}\;}^{0}}\times 100$$
where the corrosion current densities in the absence and presence of inhibitor are denoted by $j_{\mathrm{corr}\;}^{0}$ and $j_{\mathrm{corr}\;}$, correspondingly.

The EIS experiments were executed at 30 ± 2  °C in a 0.01 Hz to 10^5^ Hz frequency range and 0.01 V amplitude of sinusoidal potential perturbation. The inhibition effectiveness from the EIS experiment was estimated from Equation (4) given below [85]:(14)$$\eta_{\mathrm{eis}}\%=\frac{R_{ct}-R_{ct}^{{}^{\circ}}}{R_{ct}}\times 100\;$$

$R_{ct}\;$ and $R_{ct}^{{}^{\circ}}\;$ signify the charge transfer resistances for the blank and inhibited solutions, respectively.

### 3.5. Surface Morphology Study

The AFM, SEM, and contact angle analyses were done to approve the development of the adsorbed inhibitor layer on the metal surface by using the equipment Bruker multimode 8 (tapping mode with Nano Scope 1.5 software), JEOL JSM-7610F with EDS: INCA, OXFORD, and DSA100 Drop Shape Analyzer, respectively.

### 3.6. Theoretical Study

Density functional theory (DFT) and Monte Carlo (MC) simulations were performed on CMEA to further validate the findings obtained via the experiment. The DFT calculations were executed employing the Dmol3 (B3LYP/ DND/ COSMO (water) model) module incorporated into the Biovia Materials studio programme [64,86,87]. MC and MD simulations were achieved to study the adsorption behaviour and interaction between the CMEA and the Fe (110) surface employing “Materials Studio 8.0,” advanced by Accelrys, Inc. A three-dimensional box (27.306127, 27.306127, 20.29536) with periodic boundary conditions was used for the simulation procedure to mimic the representative portion of the interface free from any extraneous border impact. Before the simulation, the Fe (110) plane was split from the iron crystal and the surface was modified. A supercell was developed by extending Fe (110). There was constructed a 40-vacuum layer above the Fe (110) plane. CMEA molecules were also optimized. The adsorption of CMEA on Fe (110) was simulated at 298 K using Andersen’s thermostat, NVT, a simulation duration, and a time step of 0.8 ns and 1 fs, respectively.

## 4. Conclusions

The corrosion inhibition behaviour of coco monoethanolamide (CMEA), a non-ionic surfactant based on coconut, against MS in 1 M HCl was examined utilizing both experimental and computational methodologies. The CMEA displayed an excellent inhibition efficacy of 97.17% at 0.6163 mM concentration of CMEA, which was close to the CMC value (0.556 mM). No further increase in the inhibition efficiency was seen beyond this concentration attributable to the saturation of the adsorbed inhibitor layer on the MS. The weight loss results were in accordance with the EIS and PDP results. The impact of temperature was examined by a WL experiment and an improvement in the inhibition effectiveness was seen with the increase in temperature depicting chemisorption. At 60 °C, the maximum inhibition effectiveness of the order of 99% was obtained at 0.6163 mM. The adsorption isotherm that fitted was the Langmuir isotherm. From the PDP outcomes, it was evident that the CMEA was a mixed inhibitor and predominantly controlled the hydrogen evolution (cathodic) reaction, as was clear from the shift in *E*_corr_ values (<85 mV). The *R*_ct_ values increased from 21.45 to 758.53 ohm cm^2^ owing to the development of an adsorbed layer of surfactant that controlled the charge transfer process. The creation of a defensive layer was validated by the SEM and AFM micrographs that depicted smooth surfaces for inhibited coupons. From the DFT studies, the energy gap value between E_HOMO_ and E_LUMO_ was evaluated to be 7.617 eV, which supported the facile electron transfer from E_HOMO_ of the inhibitor to E_LUMO_ of the MS. From the MD trajectory’s RDF results, the bond length between the inhibitor heteroatoms and Fe was less than 3.5 Å, which confirmed the inhibitor’s chemisorption behaviour. Hence, CMEA was confirmed to be a potent corrosion inhibitor.

## Figures and Tables

**Figure 1 molecules-28-01581-f001:**
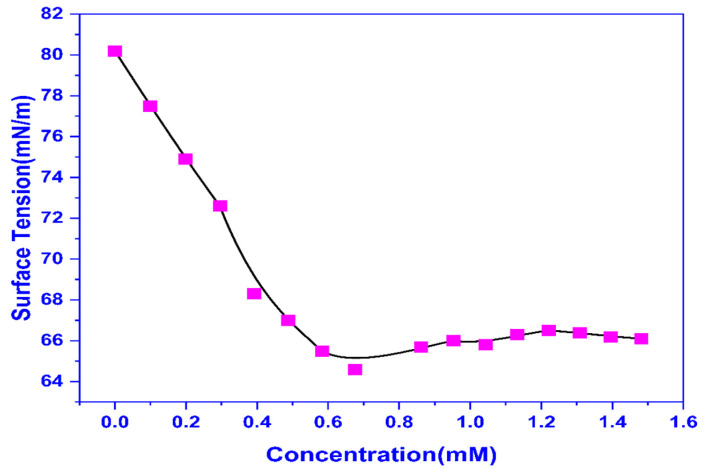
The plot of Surface tension vs. concentration.

**Figure 2 molecules-28-01581-f002:**
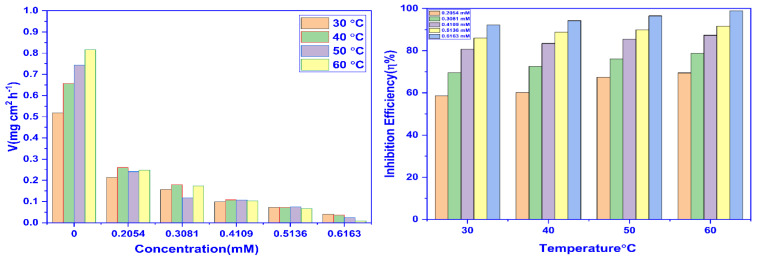
The impact of concentration (**Left**) and temperature (**Right**) on the corrosion inhibition potential of CMEA for MS in 1 M HCl.

**Figure 3 molecules-28-01581-f003:**
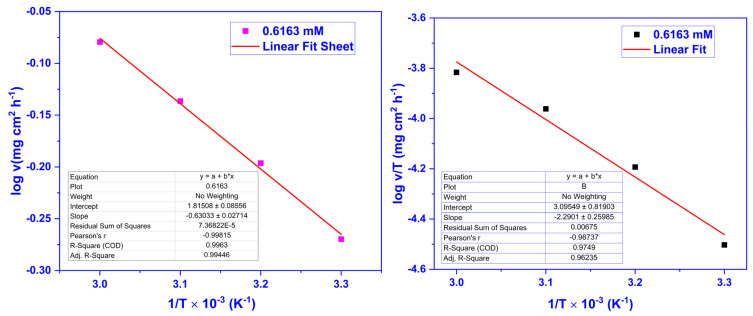
Arrhenius (**Left**) and Eyring (**Right**) plots for mild steel at 0.6163 mM dosage of CMEA at various temperatures in 1 M HCl.

**Figure 4 molecules-28-01581-f004:**
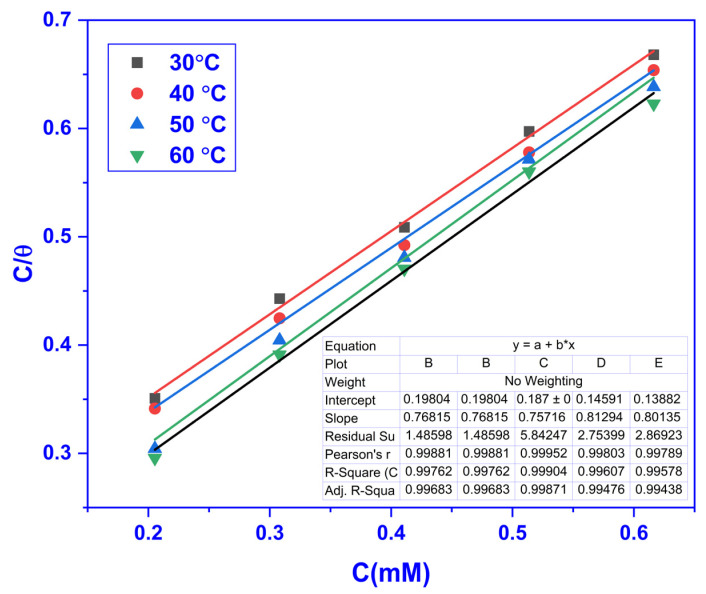
Plots showing the LAI for MS at varying CMEA doses in 1 M HCl.

**Figure 5 molecules-28-01581-f005:**
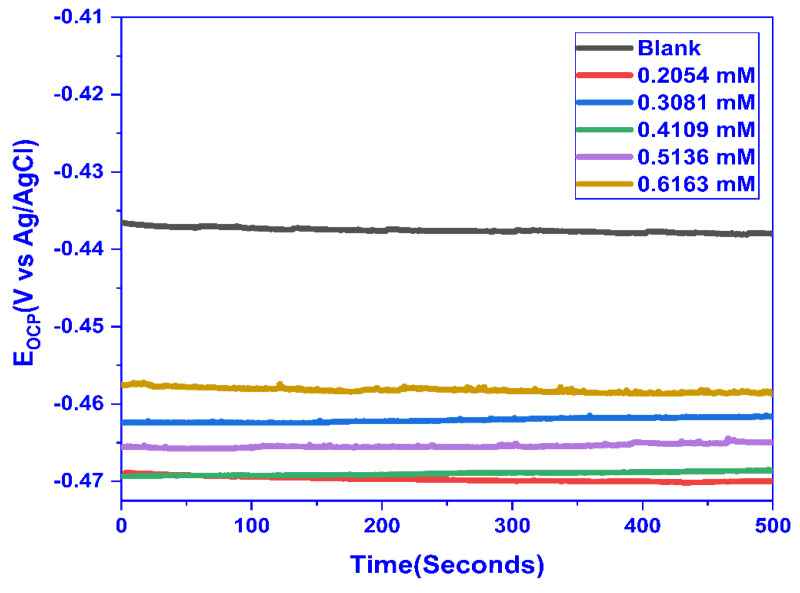
OCP vs. Time plot for MS in 1 M HCl solution with and without varying doses of CMEA.

**Figure 6 molecules-28-01581-f006:**
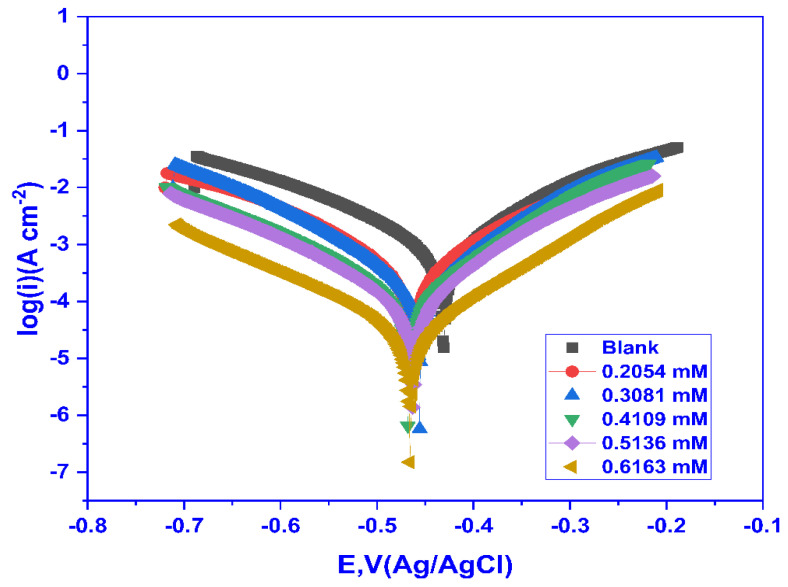
Tafel plots for mild steel for blank and varying doses of CMEA at 30 °C in 1 M HCl.

**Figure 7 molecules-28-01581-f007:**
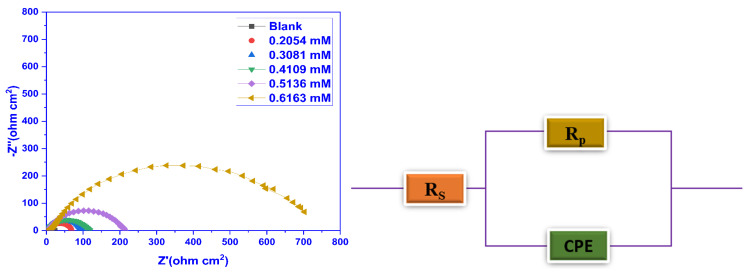
Nyquist plots for blank and different CMEA doses at 30 °C in 1 M HCl for MS (**Left**) and the electrical equivalent circuit employed to analyse the EIS data.

**Figure 8 molecules-28-01581-f008:**
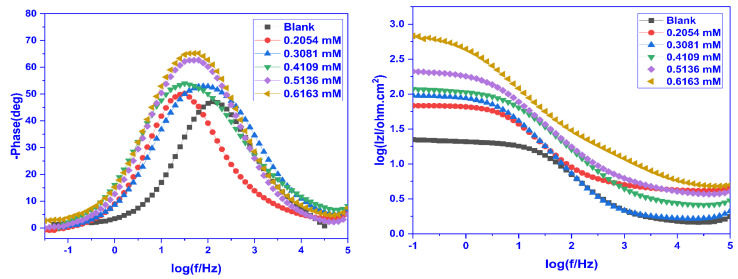
Bode phase angle (**Left**) and frequency (**Right**) plots of mild steel for a blank and inhibited solution having varying concentrations of CMEA at 30 °C in 1 M HCl.

**Figure 9 molecules-28-01581-f009:**
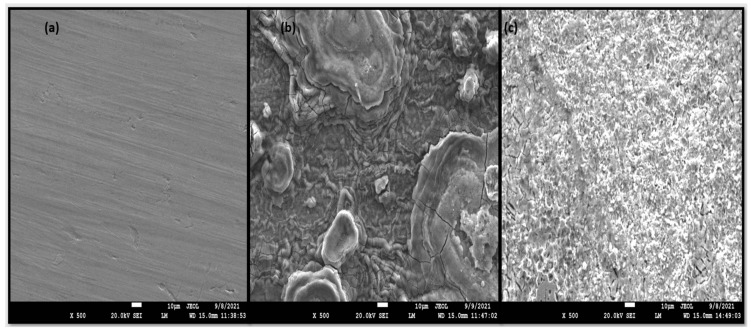
SEM photomicrographs (**a**) polished surface, after immersion in 1 M HCl for 6 h. (**b**) Blank. (**c**) CMEA.

**Figure 10 molecules-28-01581-f010:**
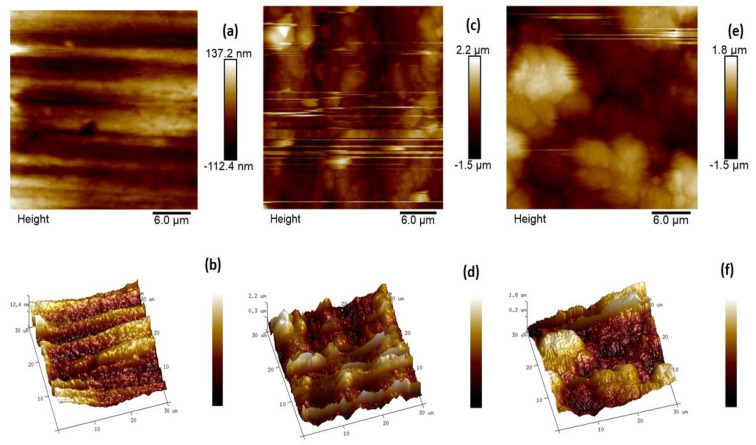
2-D and 3-D AFM micrographs for Mild steel (**a**,**b**). Polished sample. (**c**,**d**) Blank sample and (**e**,**f**) sample + 0.6163 mM CMEA in 1 M HCl.

**Figure 11 molecules-28-01581-f011:**
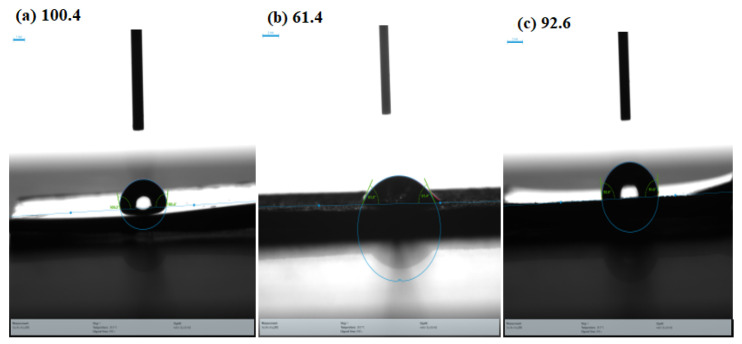
Water contact angle analysis: (**a**) polished MS, (**b**) after 6 h immersion in 1 M HCl, (**c**) after 6 h immersion in 1 M HCl + CMEA.

**Figure 12 molecules-28-01581-f012:**
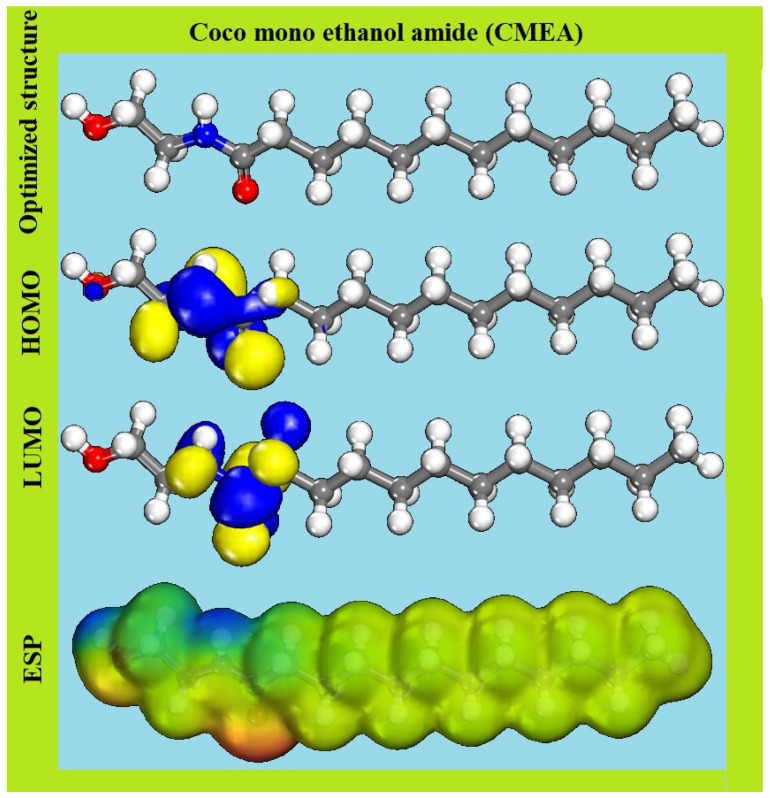
HOMO, LUMO, and ESP images of CMEA with optimised structure.

**Figure 13 molecules-28-01581-f013:**
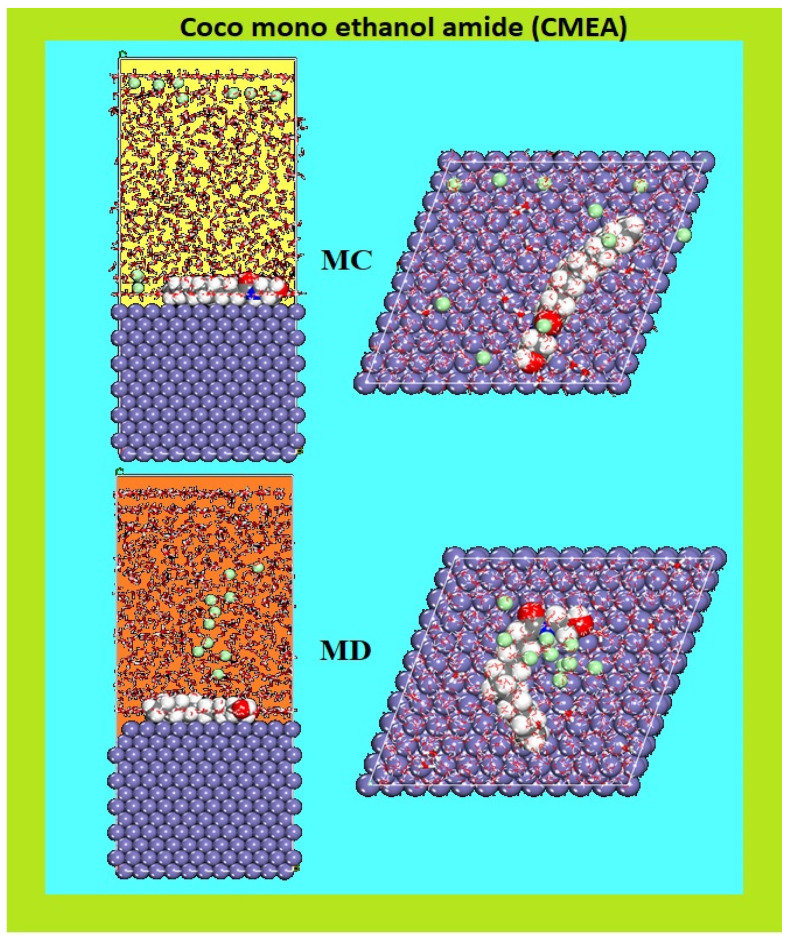
MC and MD attained from the adsorption configurations of the CMEA in the simulated corrosive media on the MS surface.

**Figure 14 molecules-28-01581-f014:**
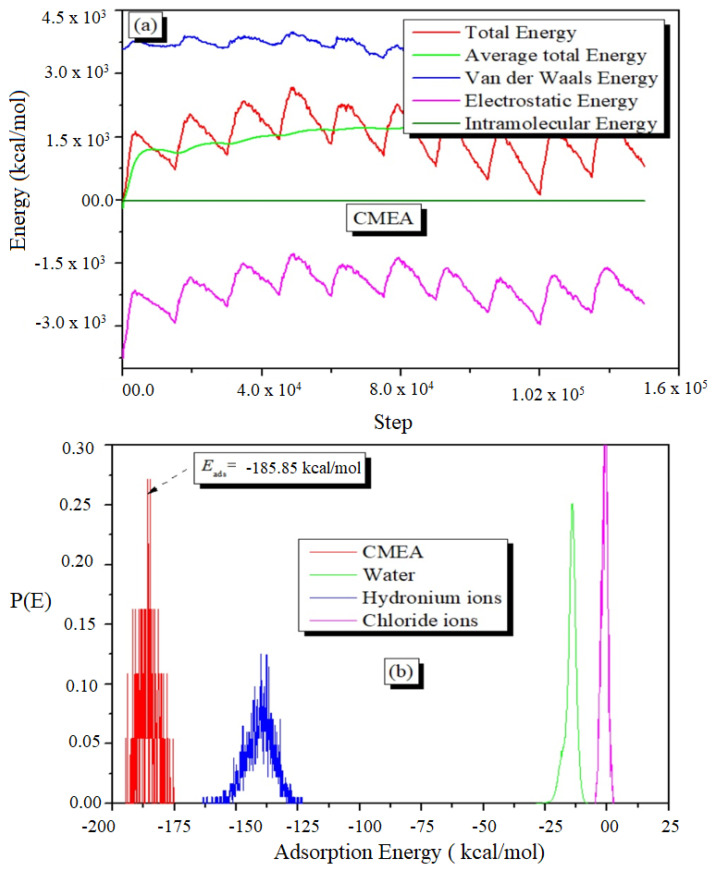
The engrossment of the various energy components throughout the MC calculations (**a**) and distribution of the CMEA inhibitor’s adsorption energies on the surface of iron (**b**).

**Figure 15 molecules-28-01581-f015:**
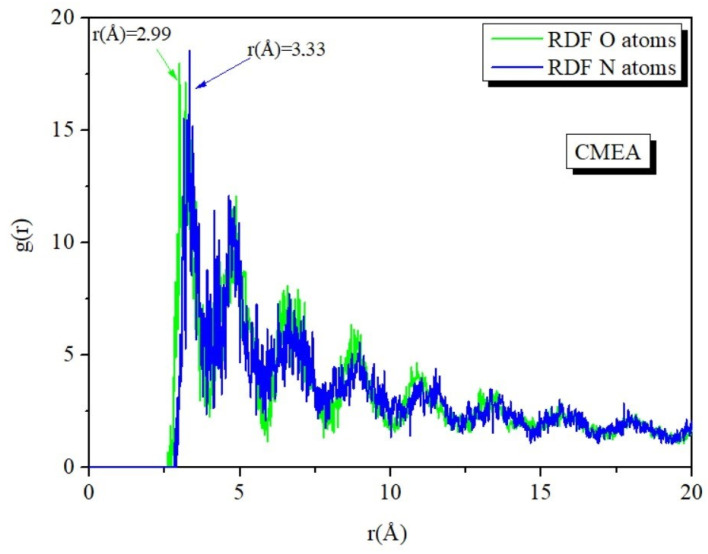
RDF O and N atoms of the CMEA in the simulated corrosive solution onto Fe surface attained by MD simulation.

**Figure 16 molecules-28-01581-f016:**
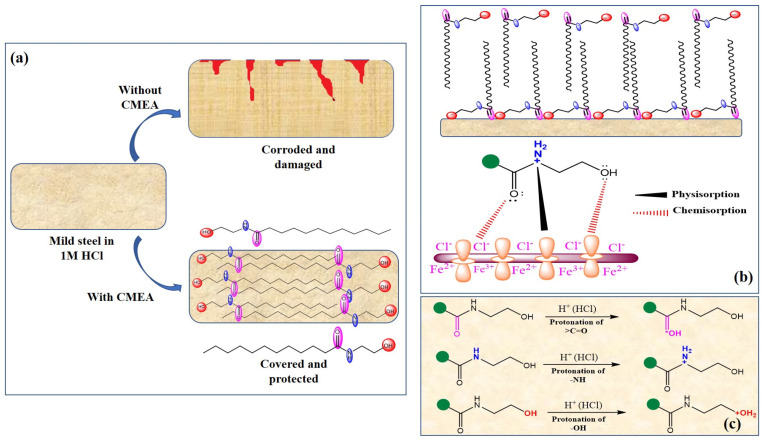
(**a**) The mechanism of corrosion inhibition. (**b**) Physisorption and chemisorption and (**c**) Protonation of different functional groups in 1 M HCl.

**Figure 17 molecules-28-01581-f017:**
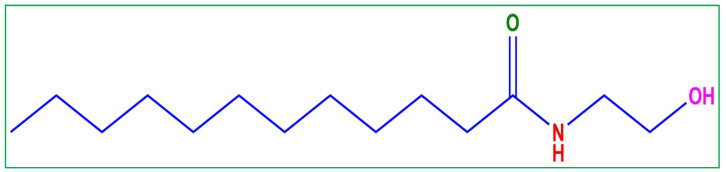
Molecular structure of coco mono ethanol amide (CMEA).

**Table 1 molecules-28-01581-t001:** WL parameters for mild steel corrosion in 1 M HCl at different doses of CMEA at 30–60°C.

Temperature (°C)	Conc. (mM)	Weight Loss (mg)	ν	Surface Coverage (θ)	η_WL_%
30 °C	Blank	55.944	0.518	-	-
	0.2054	23.112	0.214	0.5854	58.54
	0.3081	16.956	0.157	0.6956	69.56
	0.4109	10.692	0.099	0.8075	80.75
	0.5136	7.776	0.072	0.8598	85.98
	0.6163	4.32	0.040	0.9223	92.23
40 °C	Blank	70.95	0.657	-	-
	0.2054	28.188	0.261	0.6015	60.15
	0.3081	19.44	0.180	0.7253	72.53
	0.4109	11.664	0.108	0.8345	83.45
	0.5136	7.884	0.073	0.8887	88.87
	0.6163	3.996	0.037	0.9426	94.26
50 °C	Blank	80.244	0.743	-	-
	0.2054	26.028	0.241	0.6748	67.48
	0.3081	12.636	0.117	0.7617	76.17
	0.4109	11.556	0.107	0.8547	85.47
	0.5136	8.1	0.075	0.8987	89.87
	0.6163	2.7	0.025	0.9653	96.53
60 °C	Blank	88.128	0.816	-	-
	0.2054	26.892	0.249	0.6946	69.46
	0.3081	18.684	0.173	0.7876	78.76
	0.4109	11.124	0.103	0.8735	87.35
	0.5136	7.236	0.067	0.9169	91.69
	0.6163	0.864	0.008	0.9895	99.01

**Table 2 molecules-28-01581-t002:** Activation parameters obtained for MS corrosion in 1 M HCl solution in the 0.6163 mM CMEA.

Concentration (mM)	*E*_a_ KJ.mol^−1^	∆H* KJ.mol^−1^	∆S* KJ.mol^−1^.K^−1^
0.6163	40.83	43.85	−0.099

**Table 3 molecules-28-01581-t003:** Thermodynamic adsorption parameters in 1 M HCl for MS at various CMEA doses between 30 and 60 °C.

Temp. (°C)	K_ads_ × 10^4^ (M^−1^)	$\Delta G_{ads}^{0}\;\mathrm{KJ}\cdot \mathrm{mol}{}^{-1}$	$\Delta H_{ads}^{0}\;\mathrm{KJ}\cdot \mathrm{mol}{}^{-1}$	$\Delta S_{ads}^{0}\;\mathrm{KJ}\cdot \mathrm{mol}{}^{-1}\cdot K{}^{-1}$
30	0.504	−31.59	10.96	0.140
40	0.534	−32.78		0.139
50	0.685	−34.50		0.131
60	0.720	−35.70		0.140

**Table 4 molecules-28-01581-t004:** PDP parameters for MS corrosion in 1 M HCl in the absence and presence of various doses of CMEA at 30 °C.

Conc. (mM)	*E*_corr_ (V)	j_corr_ (µA cm^−2^)	*b*_a_ (V/dec)	−*b*_c_ (V/dec)	Polarization Resistance (Ω)	Corrosion Rate (mm/yr.)	η_PDP_%
Blank	−0.427	1028	0.121	0.158	28.981	11.945	-
0.2054	−0.462	410.24	0.133	0.150	63.608	5.6158	60.09
0.3081	−0.453	273.62	0.098	0.117	98.047	2.7611	73.38
0.4109	−0.467	142.56	0.092	0.140	169.83	1.6565	86.13
0.5136	−0.462	125.96	0.106	0.134	204.71	1.4636	87.74
0.6163	−0.464	29.632	0.096	0.135	828.11	0.344	97.11

**Table 5 molecules-28-01581-t005:** EIS variables for MS in 1 M HCl in lack and existence of CMEA at 30 °C.

Conc. (mM)	*R*_s_ (Ω)	*R*_ct_ (Ω cm^2^)	CPE		C_dl_ (µF cm^−2^)	η_EIS%_
			n	Y_0_ (µF cm^−2^)		
Blank	1.68	21.45	0.84	270.29	334.34	-
0.2054	5.53	63.35	0.88	336.42	317.14	66.14
0.3081	1.57	97.34	0.85	349.85	305.15	77.96
0.4109	3.17	115.31	0.77	179.16	227.64	81.39
0.5136	5.86	210.09	0.78	216.07	196.68	89.79
0.6163	7.65	758.53	0.73	225.47	181.48	97.17

**Table 6 molecules-28-01581-t006:** Results of CMEA-inhibited AFM investigation of MS.

Scheme	Polished Sample	Blank	Inhibited
Average roughness (*R*_a_)	7.31 nm	327 nm	20.4 nm
Root mean square roughness (R_q_)	8.67 nm	478 nm	24.1 nm

**Table 7 molecules-28-01581-t007:** Theoretically evaluated quantum chemical variables of the CMEA inhibitor.

Theoretical Parameters	CMEA
*E* _HOMO_	−7.203
*E* _LUMO_	0.414
∆E (*E*_LUMO_ − *E*_HOMO_)	7.617
Ionization energy (I)	7.203
Electron affinity (A)	−0.414
Dipole magnitude (µ)	7.459
Electronegativity (χ)	3.394
Global hardness (η)	3.808
Global softness (σ)	0.262
Fraction of transferred electrons (∆N)	0.473
∆E_back-donation_	−0.952

## Data Availability

The authors confirm that the data supporting the findings of this study are available within the article.

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
