# Peer review of "Coco Monoethanolamide Surfactant as a Sustainable Corrosion Inhibitor for Mild Steel: Theoretical and Experimental Investigations"

_molecules, 2023, doi:10.3390/molecules28041581_

Round 1

Reviewer 1 Report

This manuscript studied the corrosion inhibition effect of Coco Monoethanolamide Surfactant for mild stil in a acid solution (1M HCl). The corrosion inhibition of the tested inhibitors was analyzed by different methods such as eaith loss and electrochemicaltechniques. The effect of the temperature was also analysed. A computational chemical approach was used to back up the experimental results. Although the subject is interesting, the manuscript must be improved before it is suitable for publication. They are outlined below and in the PDF file:

The surface should be also analysed with other surface techniques such as XPS, RAMA, in order to support electrochemical and computational experiments.

Author Response

Reviewer’s comments: This manuscript studied the corrosion inhibition effect of Coco Monoethanolamide Surfactant for mild stil in a acid solution (1M HCl). The corrosion inhibition of the tested inhibitors was analyzed by different methods such as eaith loss and electrochemical techniques. The effect of the temperature was also analysed. A computational chemical approach was used to back up the experimental results. Although the subject is interesting, the manuscript must be improved before it is suitable for publication. They are outlined below and in the PDF file:
The surface should be also analysed with other surface techniques such as XPS, RAMA, in order to support electrochemical and computational experiments.

Author’s Response: Dear Reviewer, thank you so much for painstakingly going through our manuscript and making useful editing, comments and suggestions. We have gone through your edits, comments and suggestions and revised our manuscript accordingly. The revised areas of the manuscript are highlighted in red. Regarding your suggestion for XPS and RAMA analyses, we would like to say that currently, we don’t have these facilities in our institutions. More so, SEM and AFM studies have already been included in the manuscript to describe the adsorption mechanism of corrosion protection. However, we are considering this as a positive suggestion for our future/ next studies. In EIS studies, we have reported only Rct values as these values are the most extensively reported.  Thank you so much.

Reviewer 2 Report

The present work is concerned with studying corrosion inhabitation by coco monoethanolamide surfactant under different conditions including surfactant concentration and temperature. Also, theoretical investigations is provided. The present work in interesting and woth to be bulished in molecules journal. However, some minor changes need to be considered before publications as illustrated below:

·  It is not familiar to put figures in the introduction part of manuscript. So, it is recommended to reorient figure (1&2) in the revised manuscript.

·   The definition of some abbreviations is missing for the first time such as “WL”. So, it is recommended to make a brief check for all abbreviations in the manuscript.

·  The PD plot mostly measured at slow scan rate like 1 or 2 mV.s-1. Why authors use such relatively high scan rates 5 mV.s-1, which could affect the inhabitation efficiency results?

·   Why activation parameters only estimated at molar concentration at around 0.61 mM?

Author Response

Reviewer’s comments: The present work is concerned with studying corrosion inhabitation by coco monoethanolamide surfactant under different conditions including surfactant concentration and temperature. Also, theoretical investigations is provided. The present work in interesting and woth to be bulished in molecules journal. However, some minor changes need to be considered before publications as illustrated below:

Author’s Response: Dear reviewer, thank you so much again for going through our manuscript and making useful comments and suggestions that have greatly improved the quality of the revised manuscript. We have gone through your comments and suggestions and revised our manuscript accordingly. The revised areas of the manuscript are highlighted in red.

Reviewer’s comment 1:  It is not familiar to put figures in the introduction part of manuscript. So, it is recommended to reorient figure (1&2) in the revised manuscript.

Author’s Response: Dear reviewer, thank you so much for your valuable suggestion. Looking at the significance, we realized that Figure 1 can not be placed in the experimental and results and discussion section. Therefore, we deleted Figure 1. However, we think that Figure 2 is placed in the experimental section.

Reviewer’s comment 2:   The definition of some abbreviations is missing for the first time such as “WL”. So, it is recommended to make a brief check for all abbreviations in the manuscript.

Author’s Response: Dear reviewer, thank you so much for your valuable suggestion. The abbreviations have been carefully checked and corrected.

Reviewer’s comment 3:   The PD plot mostly measured at slow scan rate like 1 or 2 mV.s-1. Why authors use such relatively high scan rates 5 mV.s-1, which could affect the inhabitation efficiency results?

Author’s Response: Dear reviewer, thank you so much for your careful observation. Yes, we recorded the PDP plot at the scan rate of 1 mV.s-1. The 5 mV.s-1 was written by mistake. Now it has been corrected.

Reviewer’s comment 4:  Why activation parameters only estimated at molar concentration at around 0.61 mM?

Author’s Response: Dear reviewer, this is a very common practice to perform weight loss or electrochemical studies at the optimum concentration of the inhibitors for the calculations of activation and thermodynamic parameters. Therefore, we also used the optimum concentration of 0.61 mM of CMEA for the calculation of activation and thermodynamic parameters using weight loss experiments at different temperatures (30, 40, 50 and 60 °C).

Round 2

Reviewer 1 Report

The authors answered the questions satisfactorily. The manuscript can be published in the current form. Przetłumacz